# Computational analysis of US congressional speeches reveals a shift from evidence to intuition

Segun T. Aroyehun[1], Almog Simchon [2], Fabio Carrella [3], Jana Lasser [4,5], Stephan Lewandowsky [3,6] ✉ & David Garcia[1,5]

Pursuit of honest and truthful decision-making is crucial for governance and accountability in democracies. However, people sometimes take different perspectives of what it means to be honest and how to pursue truthfulness. Here we explore a continuum of perspectives from evidence-based reasoning, rooted in ascertainable facts and data, at one end, to intuitive decisions that are driven by feelings and subjective interpretations, at the other. We analyse the linguistic traces of those contrasting perspectives in congressional speeches from 1879 to 2022. We find that evidence-based language has continued to decline since the mid-1970s, together with a decline in legislative productivity. The decline was accompanied by increasing partisan polarization in Congress and rising income inequality in society. The results highlight the importance of evidence-based language in political decision-making.

Honesty and truthfulness underpin accountability, transparency and informed decision-making in democratic societies. A collective commitment to truth cultivates discourse grounded in empirical evidence and fosters social cohesion through a shared understanding of reality[1]. In many democracies, there is currently much concern about 'truth decay'[2]: the blurring of the boundary between fact and fiction[3], not only fuelling polarization but also undermining public trust in institutions[3,4].

We adopt a framework that distinguishes two rhetorical approaches with which politicians can express their pursuit of truth[5–8]. One approach, which we call evidence-based, pursues truth by relying on evidence, facts, data and other elements of external reality. An alternative approach, called intuition-based, pursues truth by relying on feelings, instincts, personal values and other elements drawn mainly from a person's internal experiences. Productive democratic discourse balances between evidence-based and intuition-based conceptions of truth. While evidence-based discourse provides a foundation for 'reasoned' debate, intuition contributes emotional and experiential dimensions that can be critical for exploring and resolving societal issues. However, although the mix of evidence-based and intuition-based

pathways to truth ranges along a continuum, exclusive reliance on intuition may prevent productive political debate because evidence and data can no longer adjudicate between competing political positions and eventually lead to an agreement. Here, we examine these developments by analysing the basic conceptions of truth that politicians deploy in political speech. We are not concerned with the truth value of individual assertions but with how the pursuit of truth is reflected in political rhetoric.

We apply computational text analysis[9,10] to measure the relative prevalence of evidence-based and intuition-based language in 145 years of speeches on the floor of the US Congress. The conceptions of truth used in congressional rhetoric are relevant to various measures of political and societal welfare. We analyse congressional rhetoric in relation to two likely drivers of democratic backsliding[11]: partisan polarization and income inequality. Polarization, characterized by growing ideological divisions and partisan animosity, undermines constructive dialogue, hampers compromise and erodes trust in political institutions, ultimately weakening democratic processes[12,13]. Previous research underscores the link between political polarization

[1]Department of Politics and Public Administration, University of Konstanz, Konstanz, Germany. [2]Department of Psychology, Ben-Gurion University of the Negev, Beersheba, Israel. [3]School of Psychological Science, University of Bristol, Bristol, UK. [4]IDea_Lab, University of Graz, Graz, Austria. [5]Complexity Science Hub, Vienna, Austria. [6]Department of Psychology, University of Potsdam, Potsdam, Germany. ✉e-mail: Stephan.Lewandowsky@bristol.ac.uk

and language use, highlighting the influence of ideological divisions on communication patterns and political behaviour[14,15]. Economic inequality is also negatively associated with various individual and social outcomes[16]. For example, individuals in environments characterized by high inequality tend to project individualistic norms onto society[17]. This fosters greater competition and reduces cooperation, which, in turn, may damage democracy[11]. Polarization can play a role in increasing inequality through lower congressional productivity[18], which could be affected by a shift from evidence-based language to intuition-based language in congressional rhetoric. This motivates our analysis of congressional rhetoric in congressional productivity, as assessed through the quantity and quality of enacted laws over time[19,20].

## Measure of evidence-based and intuition-based language

Our analysis involves 8 million congressional speech transcripts between 1879 and 2022. Details on preprocessing of the corpus can be found in the Methods. We measure the relative salience of evidence-based language over intuition-based language as the evidence-minus-intuition (EMI) score, building on a text analysis approach that combines dictionaries with word embeddings to represent documents and concepts[10] as used in previous work on political communication[21,22]. We constructed dictionaries to capture evidence-based and intuition-based language styles that underlie the two conceptions of truth (for example 'fact' and 'proof' in the evidence-based dictionary and 'guess' and 'believe' in the intuition-based dictionary; see the Methods for the full dictionaries). We adopt the approach for construction and validation of dictionaries used in[22] (see the Methods for details). Our final dictionaries consist of 49 keywords for evidence-based language and 35 keywords for intuition-based language (see Methods). We use a Word2Vec embeddings model[9] that we train on the congressional speeches. This approach converts each conception of truth into a vector representation by averaging the embeddings of the corresponding dictionary keywords. Similarly, the target text is represented as the average word embeddings of content words. We quantify the EMI score as the difference between cosine similarities of the text being analysed and the two dictionaries. A positive EMI score indicates a higher prevalence of evidence-based language (Supplementary Table 1), whereas a negative score suggests reliance on intuition-based language (Supplementary Table 2). See the Methods for further details, including validation of the EMI score against human ratings.

## Trend of EMI over time, by party and across chambers

Figure 1a shows the trend of EMI score over time, reflecting the relative prevalence of evidence-based language. EMI was high and relatively stable from 1875 through the early part of the twentieth century. Subsequently, an upward trend from the 1940s culminated in a peak in the mid-1970s. Since then, evidence-based language has been on the decline. We also include a plot showing the trends of the evidence and intuition scores in the Supplementary Note 3 (Supplementary Fig. 2).

However, notable dips in EMI also occurred during the early, stable period: the 56th Congress (from 1899 to 1901) has the historically lowest EMI score before the 1970s, closely followed by the 73rd Congress (from 1933 to 1935). These two periods align with notable historical events. In the 1890s, the USA experienced the Gilded Age, marked by rapid industrialization and economic growth, but also social unrest and increasing economic inequality. The 1930s were marked by the Great Depression during which the country faced high unemployment, widespread poverty and social upheaval. These economic and social upheavals probably influenced the language used in Congress during these periods. The profound impact of these events might have led to a greater emphasis on intuition-based language, consonant with previous research that has documented shifts in language use among individuals facing stressful situations[23] as well as among political leaders

confronted with crises[24,25]. An examination of a sample of speeches with low EMI scores in specific periods shows a tendency to focus on the crisis of the time (see Supplementary Table 3 for illustrative examples).

Focusing on the period past 1970, one striking observation is that the level of EMI has recently fallen to its historical minimum, following a decreasing linear trend that started in the peak session of 1975–1976 ($b = -0.032$, $P < 0.001$, $R^2 = 0.927$). Figure 1b illustrates the temporal trend of the EMI score for Democrats and Republicans separately. There is a strong positive correlation between the EMI scores for both parties (Pearson's $r = 0.778$, 95% confidence interval (CI) = [0.666, 0.855], $P < 0.005$). We observe some divergence between parties in the early periods. However, since the mid-70s, both parties have moved largely in the same downward direction in their rhetoric. The same pattern holds for both parties across the House and Senate (Supplementary Fig. 3). It is, however, noticeable that the EMI of Republicans dropped substantially, and more steeply than for Democrats, in the last session (2021–2022). A Mann–Whitney test shows that the difference in median EMI score (−0.435 for Democrats and −0.753 for Republicans) is significant ($P < 0.001$).

To exclude a dependence of the trend observed in Fig. 1a on topic composition of the speeches, we aggregate the EMI score by taking a macroaverage over topics such that topics have equal weighting. The results of this approach show a very similar trend (Supplementary Note 10).

To address potential concerns about semantic change over the extended timescale of this study, we perform an analysis of the stability of the meaning of dictionary keywords and also compute the EMI score using temporal embeddings. The results (Supplementary Note 11) show that the meanings of the keywords are relatively stable over time, and the trend of the EMI score computed using temporal embeddings aligns closely with Fig. 1a.

Turning to the potential correlates of the observed trends, Fig. 1c shows partisan polarization in Congress over time, measured as the difference between the first dimension of DW-NOMINATE (Dynamic Weighted NOMINAl Three-step Estimation) scores[26,27] for the two major parties averaged across the House and Senate. It is important to clarify that the political polarization indicator used in this study, DW-NOMINATE, measures polarization using voting behaviour within a legislative context. However, polarization is a complex and multifaceted concept with various definitions and indicators, including affective polarization, issue polarization and perceived polarization. Additional indicators can also be derived from computational text analysis, as well as from opinion, structural and interactional dynamics. Exploring these alternative measures is beyond the scope of the current study.

Figure 1c also includes the trend of income inequality[28] using the share of pretax income of the top 1% of the population (source: https://wid.world/). The recent decline of EMI is accompanied by a corresponding upward trend in partisan polarization in Congress and rising income inequality in society, which is statistically supported as follows.

## EMI and polarization

EMI and polarization are negatively cross-correlated (Pearson's $r = -0.615$, 95% CI = [−0.741, −0.447], $P < 0.005$), and a lagged correlation analysis shows that lag zero has the highest correlation (Supplementary Fig. 6a). Supplementary Fig. 5 also depicts the relationship. When included in lagged regression models, EMI does not explain a significant amount of the empirical variance of polarization, but polarization has a significant coefficient in the EMI model ($b = -0.15$, 95% CI = [−0.29, −0.01], $P < 0.05$). Refer to the Methods section for details of the regression results.

## EMI and income inequality

EMI values are informative of future inequality. Figure 2 shows the historical values of inequality as a function of EMI in the previous

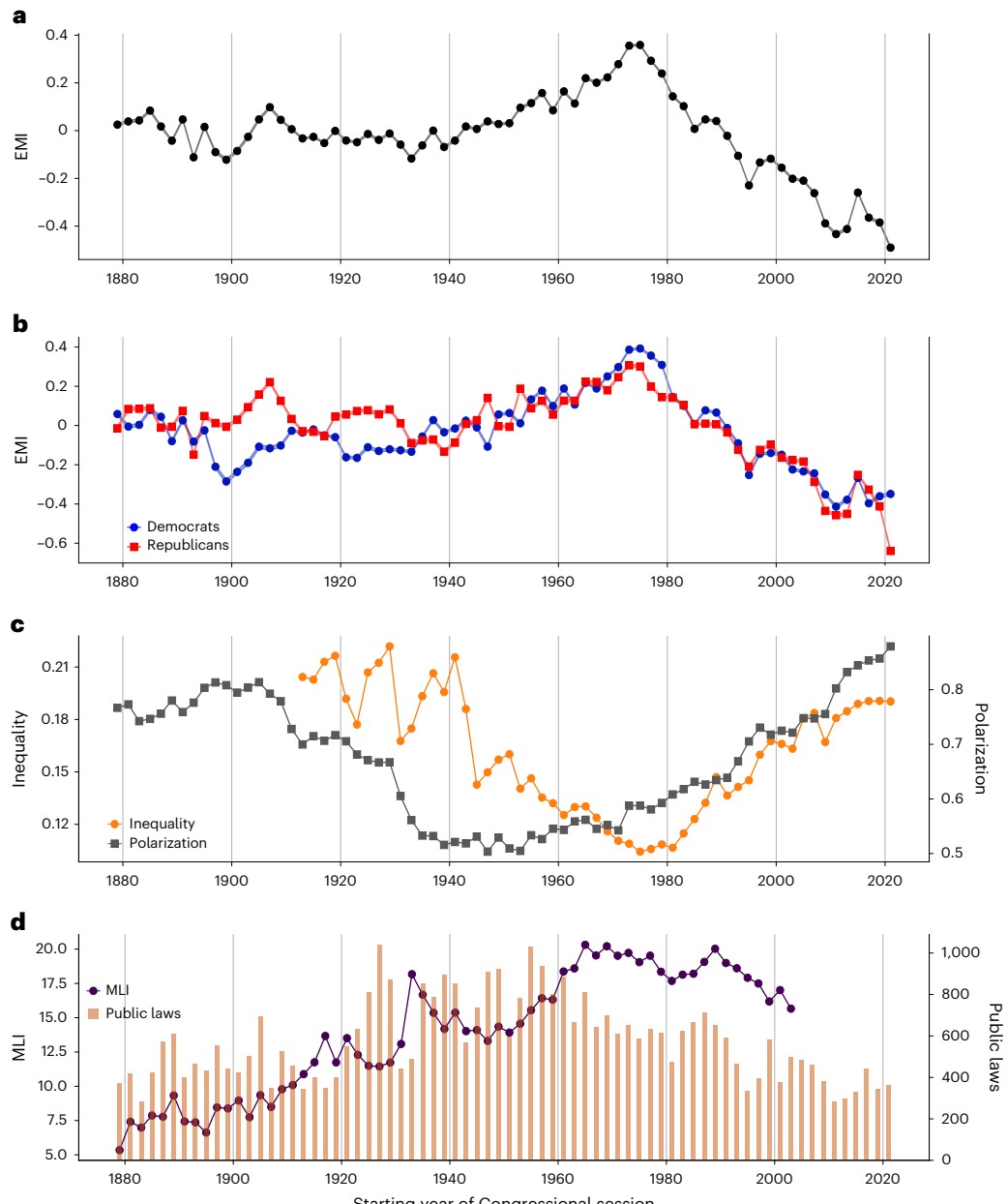

**Fig. 1 | Trend of the EMI score, congressional polarization, inequality, and legislative productivity. a–d**, Time series of the EMI score in each congressional session between 1879 and 2022 (**a**), EMI scores separated by party (**b**), congressional polarization and inequality (**c**) and congressional productivity, measured as the MLI and the number of public laws passed by each session (**d**). We compute bootstrapping 95% CIs for EMI with 10,000 samples, which may appear too small to be visible owing to the large sample size.

session, that is, the previous two years (Pearson's $r = -0.948$, 95% CI $= [-0.973, -0.902]$, $P < 0.001$). A lagged correlation analysis shows that the strongest correlations appear when EMI precedes inequality (Supplementary Fig. 6b).

This is buttressed by a lagged regression model including the level of inequality, polarization and EMI from the previous session, as well as their interaction. The results of that fit in comparison with an autoregressive (AR) model reveal a negative coefficient of EMI with inequality 2 years later ($b = -0.11$, 95% CI $[-0.20, -0.02]$). Details of these models are in the Methods section. The interaction with polarization is not significant and weak enough for the slope of EMI to stay negative (Supplementary Fig. 7). These regression results are robust to other specifications of the analysis, for example, when using the Gini index instead of the top 1% share of income (restricted to the time since full income data became available), when using all available data since 1912 and when considering a longer lag for polarization (Supplementary Table 4).

## Relationship between EMI and congressional productivity

Evidence-based language can be a tool to identify factual constraints for Congress to formulate legislation, which often requires some form of bipartisan agreement. We examine the relationship between EMI and congressional productivity as measured by three indicators. First is the major legislation index (MLI)[19] which measures the productivity of Congress in terms of important legislation. Second is the legislative productivity index (LPI)[19], which combines assessments of important legislation and number of laws enacted. Third is the count of the number

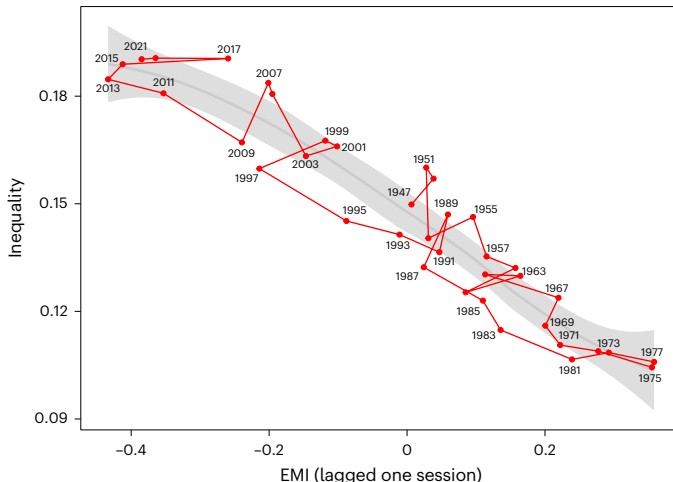

**Fig. 2 | Inequality measured as the share of income of the top 1% versus the EMI score in the previous legislature.** The shaded area shows the 95% CI of a locally estimated scatterplot smoothing (LOESS) fit, and labels indicate the year corresponding to the inequality measurement.

of laws passed by each session of Congress[20] without considering their significance. Previous research analyses congressional productivity as a function of polarization, party composition in the legislature and executive branch[19] and public mood towards more regulation as measured in surveys[29]. From these indicators, polarization and public mood towards regulation are the most important predictors, explaining a significant amount of the variance of productivity over time[19].

Figure 3 shows the relationship between all three congressional productivity metrics and EMI measured in the same session. All three cases have positive and significant correlations (MLI: Pearson's $r = 0.454$, 95% CI = [0.09, 0.711], $P < 0.05$; LPI: Pearson's $r = 0.836$, 95% CI = [0.667, 0.923], $P < 0.001$; log-transformed number of laws: Pearson's $r = 0.796$, 95% CI = [0.633, 0.891], $P < 0.001$). However, polarization and public mood about regulation play an important role in congressional productivity, which is shown by the colour of the plotting symbols in Fig. 3. Points representing high public mood (blue) tend to lie above the regression line, and points with low public mood (red) tend to lie below. For that reason, we fitted the base models of ref. 19 and tested if adding the EMI of a session has a positive association with the LPI. The results (see Methods for details) reveal that, after controlling for known correlates in productivity and for an interaction between polarization and EMI, the coefficient of EMI is positive and significant for MLI ($b = 0.67$, 95% CI = [0.14, 1.20], $P < 0.05$) and LPI ($b = 0.83$, 95% CI = [0.40, 1.26], $P < 0.05$), and positive for the number of laws but not statistically significant ($P < 0.1$). We see this as an indication that EMI plays a role in congressional productivity, with the association being more salient when considering major legislation in comparison with minor laws where parliamentary debate might not play a bigger role.

## Discussion and conclusion

We introduce an approach for quantifying the conception of truth that members of Congress embrace and deploy in their rhetoric. Using embedded dictionaries in conjunction with embedding of congressional speeches, we calculate and validate the EMI score from transcripts of congressional speeches spanning the years 1879–2022. The EMI score reflects the prevalence of evidence-based language when positive and intuition-based language when negative. We study the temporal trends of the EMI score and investigate its relationships with measures of polarization and inequality as well as congressional productivity.

We find that EMI shows a pattern of relative stability until the 1940s, which is followed by a clear upward trajectory that reached a maximum in the 1970s. Since then, EMI trends downwards, indicating a decline in the prevalence of evidence-based language for both parties. The degree of synchronization in the linguistic styles used by both Democrats and Republicans during this period points to their alignment around messaging strategies[30].

We examine the decline in EMI in relation to three outcome variables that are indicative of democratic health and find a concerning association in all cases: a decline in evidence-based language is associated with increasing polarization and increasing income inequality but decreased congressional productivity. The temporal sequence of those trends differs between variables. For polarization, the strongest association with EMI is greatest at lag zero, and we find that polarization is a significant predictor of EMI, but not vice versa. This suggests that polarization and politicians' rhetoric evolve in tandem. By contrast, EMI precedes shifts in income inequality, such that a stronger emphasis on evidence-based reasoning is associated with subsequent reduction in income inequality whereas greater reliance on intuition seems to be associated with the persistence of existing social disparities. This finding aligns with existing research on language and social inequality[31], which underscores how language patterns have consequences for understanding social issues and may either promote or inhibit necessary changes. Intuition-based language may help to explain the relationship between polarization and inequality, as it is linked to legislative inaction and can hinder policies that address income inequality through redistribution[18].

Finally, the association of evidence-based language with congressional productivity is again contemporaneous. In the Habermasian view of communicative action[32], evidence-based language serves as the foundation for 'reasoned' debates and can steer discussions away from personal and political hostilities. In this communicative process, evidence-based language serves as a tool to establish a shared understanding of the state of the world and contributes to the formulation of well-informed decisions. The positive correlation that we observe in our study between the EMI score and legislative productivity (in terms of quality and quantity) is in line with this viewpoint.

The observed patterns in congressional language are the result of a complex interplay of various factors, some of which are unique to the political and societal context of the USA. One contributing factor to these patterns is the control exerted by party leadership over who speaks on the congressional floor[33], potentially shaping the content and tone of speeches. This control mechanism is likely to influence the language used by congressional members in aligning with the strategic objectives of their party. In addition to the influence exerted by party leaders, members of Congress may find themselves compelled to cater to their base, encompassing constituents, donors and lobbyists, particularly in a highly polarized environment driven by partisanship[34].

Modifications to congressional rules and procedures, particularly around the length of debates, can influence the breadth and depth of discussions on the congressional floor. For example, the introduction of the 'cloture' rule in the Senate in 1917 provided a mechanism to limit debate time and expedite legislative processes. Before this, there was no formal method to end a debate or force a vote on an issue, which allowed extended deliberations. While such rules may improve efficiency, they can also shorten discussions and potentially limit the richness of legislative debates. The evolving nature of congressional rules and procedures can influence the characteristics of discourse on the congressional floor over time.

Presidents have increasingly sought to expand their powers, often justified by their role as commander-in-chief, particularly during crises or in an attempt to unilaterally advance their policy agendas[35]. Mechanisms such as executive orders and the creation of administrative agencies under presidential control have facilitated this expansion. While some of these actions are supported by congressional authorization, the steady accumulation of executive power may have implications for the legislative branch. This expansion may limit the sphere of influence

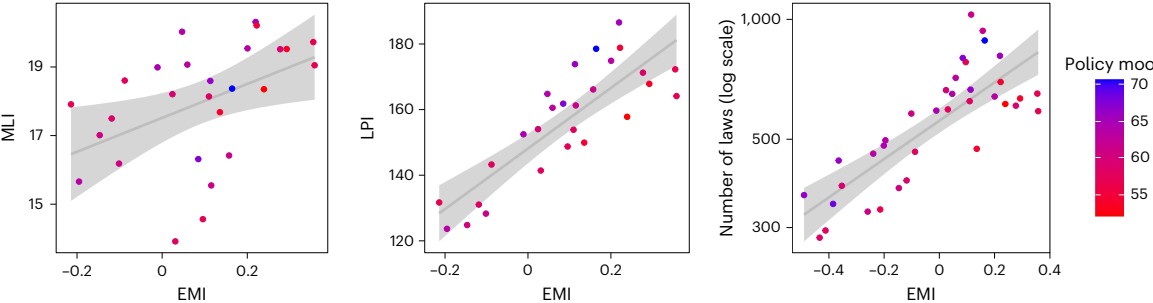

**Fig. 3 | EMI score versus congressional productivity measured as MLI (left), LPI (centre) and log-transformed number of laws (right).** Points are coloured according to public mood towards regulation during the legislative period. The grey lines represent linear regression models of each productivity variable as a function of EMI alone, and the shaded areas indicate the 95% CIs for the regression fits.

of Congress, potentially reducing its role to rubber-stamping presidential initiatives. Conversely, it can also lead to tensions and heightened oversight efforts by Congress on activities of the executive branch and agencies. The balance of power between the executive and legislative branches can shape the nature and focus of congressional discourse.

Furthermore, the impact of media on politicians, particularly their adoption of media logic[36], introduces an additional dimension to the nature of political representation. This influence could be amplified by the live coverage of proceedings through the C-SPAN (first introduced in the House in 1979 and then in the Senate in 1986). In an era characterized by increasing polarization, politicians might find themselves driven to embrace a perpetual campaign style of representation[37], transforming congressional speeches into orchestrated performances aimed at capturing media attention. Consequently, this shift may result in a reduced focus on meaningful intellectual discourse and nuanced policy discussions within the legislative body. This interpretation meshes well with a recent analysis of the Twitter/X communications of US Congress members from 2011 to 2022, which similarly differentiated between evidence-based 'fact-speaking' and authentic 'belief-speaking' as alternative expressions of honesty[22]. That study discovered an association between the prevalence of authentic belief-speaking and a decrease in the quality of shared sources in tweets, particularly among Republicans. This suggests a potential link between belief-based language and the dissemination of low-quality information to the public.

The findings presented in this study highlight important correlational associations. The absence of causal evidence underscores the need for future research to further establish definitive causal relationships.

We have highlighted concerning trends in Congress where evidence-based language is declining and partisan polarization is increasing. The decline in the quality and quantity of legislative output at a time of multiple global crises should be of concern. On a more positive note, understanding the complex relationship between the language of political discourse and partisan polarization points to avenues for interventions focused on fostering more constructive and productive debate. Initiatives such as those promoting collaboration and communication across partisan boundaries[38] can contribute to rebuilding a more robust democratic discourse. Ultimately, the challenge lies in having a Congress (and, by extension, a deliberative public) where truth is valued, polarization is in check and legislative outcomes reflect the diverse needs of the citizens.

## Methods

We initially rely on the dataset compiled by Gentzkow et al.[39] and supplement it with recent data obtained by accessing the congressional records' website using an automated script[40]. The dataset includes essential metadata such as speaker information (including party) and dates. The dataset consists of 14,153,443 speeches spanning the congressional sessions of 1879–2022. To ensure the quality of our dataset,

we use a number of preprocessing steps. First, we remove procedural speeches. Procedural speeches are speeches delivered by members of Congress that mainly deal with the rules and procedures that govern legislative proceedings. These may include discussions on amendments to rules, requests for unanimous consent, or the announcement of votes. We train three classifiers, following the methodology outlined by Card et al.[41], to identify procedural speeches. We remove procedural speeches by using a majority vote ensemble of the classifiers.

In general, the congressional record is of high quality. However, in the earlier years, it contains some instances of optical character recognition errors that result in unintelligible content (for example, in the rendering of a table). To mitigate the potential noise from lengthy speeches that consist mainly of lists of names or numbers, we use a filtering mechanism. This filter evaluates the ratio of common (top 100) English words (for example, 'the', 'and' and 'is') to the total token length of a speech. We set a threshold of 0.05, ensuring that speeches with substantive content are retained for further analysis. We keep speeches that are attributed to members of the two major parties. We filter out speeches with fewer than 11 tokens and remove duplicate entries. Our final dataset consists of 8,435,769 speeches with an average length of approximately 199 tokens. Speeches made by Democrats account for 53% of the dataset and 47% of speeches are by Republicans. Supplementary Fig. 1 shows the number of speeches for each congressional session across both chambers (House and Senate) from 1879 to 2022. The number of speeches for each session varies. Nevertheless, there is a substantial amount of speeches available, with at least 35,000 speeches for each session, to enable a reliable analysis. To facilitate further analysis, we split longer speeches (consisting of more than 150 tokens) into chunks of approximately 150 tokens each. We set a minimum chunk size of 50 tokens, such that a chunk smaller than the minimum size is merged with the immediately preceding chunk.

### Steps for the construction and validation of EMI score

**List of keywords.** We start with seed keywords, one for each conception of truth, generated by the researchers involved in this work. The goal is to capture linguistic cues that may signal the pursuit of truth in a speaker. Initial keywords for evidence-based language include 'reality', 'assess', 'examine', 'evidence', 'fact', 'truth' and 'proof'. For intuition-based language, the initial keywords include 'believe', 'opinion', 'consider', 'feel', 'intuition' and 'common sense'. We expand these lists computationally using a combination of fastText embeddings[42] and Colexification networks[43]. Using fastText embeddings, we expand the seed words by including those words with a cosine similarity score above 0.75. Colexification networks connect words within a language on the basis of their common translations across other languages, thus identifying words that express related concepts. For instance, the words 'air' and 'breath' are considered colexifications because they translate into the same word in multiple languages. Incorporating colexification networks into lexicon expansion results in word lists

**Table 1 | Dictionary for evidence-based and intuition-based language**

| Evidence keywords | | | | |
|---|---|---|---|---|
| accurate | exact | intelligence | precise | search |
| analyse | examination | investigate | procedure | show |
| analysis | examine | investigation | process | statistics |
| correct | expert | knowledge | proof | study |
| correction | explore | lab | question | trial |
| data | fact | learn | read | real |
| dossier | find | logic | reason | true |
| education | findings | logical | research | truth |
| evidence | information | method | science | truthful |
| evident | inquiry | pinpoint | scientific | |
| **Intuition keywords** | | | | |
| advice | doubt | mislead | suggestion | belief |
| fake | mistaken | suspicion | believe | fake news |
| mistrust | view | bogus | feeling | opinion |
| viewpoint | common sense | genuine | perspective | wrong |
| deceive | guess | phony | deception | gut |
| point of view | dishonest | instinct | propaganda | dishonesty |
| intuition | sense | distrust | lie | suggest |

with a better trade-off between precision and recall compared with methods relying solely on word embeddings[44]. We filter the expanded lists by removing duplicates and terms appearing in both categories. In addition, we retain only one variant of lemma inflections (for example, 'investigate', 'investigates' and 'investigated'). Following the same approach used in ref. 22, we then recruited participants on Prolific to rate each keyword on their representativeness on two scales, one for evidence-based and one for intuition-based language. We then keep only words rated as statistically more representative for their respective construct than the other. We received ethics approval from the University of Bristol for the validation of words included in the dictionaries. Informed consent was obtained from all participants before their participation in the annotation task. The annotation task was performed in accordance with relevant ethical guidelines and regulations.

Our final dictionaries consist of 49 keywords for evidence-based language and 35 keywords for intuition-based language (Table 1). The difference in the number of keywords is not a concern in our approach, because we use the distributed dictionary representation method[10], which effectively normalizes the impact of varying keyword counts by representing each dictionary with a single vector. This ensures a consistent measure of evidence-based and intuition-based language, enabling meaningful comparisons across both constructs.

**Computation of EMI score.** In our methodology, we start by training 300-dimensional word embeddings using the Word2Vec[9] algorithm on the corpus of congressional speeches. We use the Gensim library[45]. Word2Vec is an algorithm that generates dense vector representations of words, known as word embeddings. The rationale behind using Word2Vec lies in its ability to capture semantic relationships among words by representing them in a continuous vector space. This algorithm learns to predict the context of a word on the basis of its surrounding words or vice versa. The resulting word embeddings encode semantic similarities, making them valuable for computational analysis of language.

Following this, we compute a representation for the concepts of interest by averaging the word embeddings for the relevant keywords in the respective dictionaries for evidence-based and intuition-based language. For a given text, we compute its representation by taking the average of the word embeddings for its content words. This representation allows a graded measure of relatedness to each construct as we can calculate the cosine similarity between each construct representation and the representation of a given target text that is computed in the same manner.

To generate the representations and compute cosine similarities, we use the sentence-transformers library[46], leveraging our trained Word2Vec model. This approach offers efficiency and effectiveness in capturing the semantic content of textual data. This set-up allows us to obtain textual embeddings with minimal computational resources and ensures the scalability of our analysis.

To address variations in the length of speeches, we perform length adjustments for the cosine similarities. This involves binning the similarities by length and subtracting the mean similarity within each bin from the cosine similarity of each instance. Subsequently, we apply a $Z$-transform to the cosine similarities to derive the evidence and intuition scores. Finally, we obtain the EMI score by subtracting the intuition score from the evidence score. A positive EMI score indicates a higher prevalence of evidence-based language, whereas a negative score suggests a reliance on intuition-based language. Supplementary Tables 1 and 2 contain illustrative examples of speeches with positive and negative EMI scores, respectively. For further analysis, we take the mean of the EMI score per 2-year period, corresponding to the typical duration of congressional sessions. For completeness, we also include a plot of the trends for each of the component scores (Supplementary Fig. 2).

Given the extended timescale of this study, concerns may arise about semantic change and the possibility that the embeddings model relies mostly on more recent data. To address these concerns, we train temporal embeddings on two-decade slices of the speeches and downsample the data to ensure comparable token counts across time periods. We conduct an analysis on the stability of the semantics of dictionary keywords and compute the EMI score using the resulting temporal embeddings. The results (Supplementary Note 11) show that the semantics of the keywords remain relatively stable over time and that the trend of the EMI score is qualitatively similar to Fig. 1a.

**Validation of EMI score over time.** We split the EMI score into four bins per decade. We sample five (four for the most recent decade) (quasi)sentences from each bin per party (Democrats versus Republicans) and decade, resulting in a sample of size 592. We ask participants on Prolific to rate to what extent a given text is evidence-based and intuition-based (or evidence-free) on two Likert scales ranging from 1 to 7. We received an ethics review exemption from the University of Konstanz ethics review board for the annotation task used to validate the EMI score. Informed consent was obtained from all participants before their participation in the annotation task. The annotation task was performed in accordance with relevant ethical guidelines and regulations. Each text has at least five ratings. We collected a total of 4,563 human ratings from 156 participants. The average number of ratings provided by each participant is 29 (with a minimum of 11 and maximum of 30). As the average of the ratings for each scale are negatively correlated at the document level ($-0.85$, $P < 0.001$), we derive human judgement by assigning a label of evidence-based if the average evidence-based rating is greater than the average intuition-based rating; otherwise, we classify the item as intuition-based. Annotators have relatively high levels of agreement. Sampling five annotations at random for each text, the intraclass correlation for the mean of the difference between the evidence-based and intuition-based scales is 0.714 (95% CI = [0.675, 0.749]).

We calculate the area under the curve (AUC) of the receiver operating characteristics curve as the evaluation metric following previous work that used classification metrics[21,22]. We compute the AUC score per decade and for all samples. The AUC is a measure of the reliability

**Table 2 | AUC per decade and overall AUC computed on the full sample without temporal split**

| Decade starting | AUC | Number of speeches |
|---|---|---|
| 1879 | 0.81 | 40 |
| 1889 | 0.60 | 40 |
| 1899 | 0.61 | 40 |
| 1909 | 0.82 | 40 |
| 1919 | 0.61 | 40 |
| 1929 | 0.83 | 40 |
| 1939 | 0.82 | 40 |
| 1949 | 0.82 | 40 |
| 1959 | 0.93 | 40 |
| 1969 | 0.93 | 40 |
| 1979 | 0.90 | 40 |
| 1989 | 0.94 | 40 |
| 1999 | 0.93 | 40 |
| 2009 | 0.83 | 40 |
| 2019 | 0.74 | 32 |
| Overall | 0.79 | 592 |

of our computed EMI score, with a score of 1 indicating perfect accuracy and 0.5 representing performance equivalent to random chance. Our method achieves an overall AUC of 0.79 across decades, ranging from 0.60 to 0.94 (Table 2). Compared with the random baseline AUC of 0.5, our method demonstrates acceptable to excellent discrimination levels[47].

**Example speeches in periods with low overall EMI score.** Supplementary Table 3 presents examples of speeches with low EMI score (in the bottom 1%) in periods with overall low EMI scores in Fig. 1a. Consistent with previous research[23–25] that highlighted changes in the language of individuals and political leaders during crises, these examples suggest a tendency for discussions about the crisis of the time to rely more on intuition-based language rather than evidence-based language.

**EMI in the House and Senate by party.** Supplementary Fig. 3 shows the trend of EMI by party in both chambers of the US Congress over time. The trends follow a similar pattern to the one observed for the overall EMI score in the main text in Fig. 1a,b.

**Statistical analysis of the trends in EMI, polarization and inequality**
We fit time series as linear regression models that include lagged dependent variables to consider autocorrelation. For each time series, we fit AR models with increasing lags up to a point in which the quality of models does not improve with additional lags. In all cases we report, inclusion of one lag generated the best univariate AR model. We next extend these models with other variables including EMI and other covariates. We measure variance inflation factors (VIFs) of the independent variables of the models and include interaction terms when any of the covariates, excluding the lagged dependent variable, has a VIF above 10. After fitting a model specified in this way, we measure standard errors and P values with a heteroskedasticity and autocorrelation (HAC)-adjusted estimator. We assess the stationarity of residuals with augmented Dickey–Fuller (ADF) tests and Kwiatkowski–Phillips–Schmidt–Shin (KPSS) tests, and the normality of residual distributions with Jarque–Bera (JB) tests. Models generally passed these regression diagnostics, being able to reject the null hypothesis of the ADF test at a 0.05 level and failing to reject the null of the KPSS and JB tests at a

0.1 level. We report here any relevant cases where those diagnostics are different.

In our primary analysis of inequality, we consider the fraction of income of the top 1% from 1944, which is the year when tax declaration exemption rules qualitatively changed and led to more reliable inequality metrics[48]. We assessed the robustness of our results with alternative specifications, namely using the Gini index (from https://www.census.gov/data/tables/time-series/demo/income-poverty/historical-income-inequality.html) in the same period and using the full record of the share of income of the top 1% from 1912.

We add one more specification to robustly test how the role of polarization influences our results about EMI and inequality. A lagged correlation analysis between inequality and polarization indicates that the correlation between these two is strongest when considering a lag of eight legislative sessions (Supplementary Fig. 6c). To consider this longer lag, we fitted an additional regression model of inequality with EMI and the previous value of inequality, but with the value of polarization eight sessions prior. Results of this fit are reported in Supplementary Table 4.

**EMI negative trend.** The session with the highest EMI score is 1975–1976, with a score of 0.358, closely followed by the previous session with an EMI score of 0.355 but substantially higher than the mean session, which has a slightly negative EMI of −0.017. The peak EMI is more than two standard deviations (s.d.) above the mean of the historical distribution (s.d. 0.174). From that peak, a downward trend is noticeable and is confirmed by a linear regression model of the form

$$\text{EMI}(t) = a + b \times t.$$

The fit has an intercept $a = 0.258$ and a slope $b = -0.032$, both with $P < 0.001$. The model has $R^2 = 0.927$, and the fit can be seen in Supplementary Fig. 4. This is further corroborated by breakpoint analyses (Supplementary Note 12) that identified the session 1973–1974 (the session before 1975–1976) as a breakpoint.

**EMI and polarization.** To measure partisan polarization in Congress, we use the first dimension of the DW-NOMINATE score[26], which measures the ideological position of members of Congress derived from their roll-call votes. The difference in aggregate score for the two major parties reflects the extent to which they differ ideologically. A higher difference in the first dimension indicates a greater ideological distance or polarization between the parties. A lower difference suggests a closer alignment in their ideological positions. We use the DW-NOMINATE data from Voteview[27], which offers a comprehensive and widely used resource for studying the ideological landscape and partisan dynamics within the US Congress.

To understand the relationship between EMI and polarization (scatter plot in Supplementary Fig. 5), we fitted lagged regression models of the form

$$\text{EMI}(t) = a + b \times \text{EMI}(t-1) + c \times \text{Pol}(t-1)$$
$$\text{Pol}(t) = a + b \times \text{Pol}(t-1) + c \times \text{EMI}(t-1)$$

and compared them with AR models ignoring the other variable. Results of the fits (Table 3) show that polarization does not have a significant coefficient in the EMI model and that the polarization model has a significant negative coefficient for EMI, but of small magnitude compared with the AR coefficient of EMI. A KPSS test of residuals in this model rejects the null hypothesis ($P = 0.036$), but an ADF test also rejects the null ($P = 0.02$). While residuals deviate a bit from being stationary, we use HAC covariance matrix estimation and residuals do not significantly deviate from normality, as a JB test is not significant ($P = 0.645$).

**Table 3 | Models of temporal association between polarization (Pol) and EMI**

| | EMI | EMI | Pol | Pol |
|---|---|---|---|---|
| EMI($t$−1) | **0.98** | **0.92** | | |
| | [0.86,1.10] | [0.79, 1.05] | | |
| | $P=9.52\times10^{-26}$ | $P=6.66\times10^{-22}$ | | |
| Pol | | **−0.15** | | |
| | | [−0.29, −0.01] | | |
| | | $P=3.29\times10^{-2}$ | | |
| Pol($t$−1) | | | **1.00** | **0.97** |
| | | | [0.96, 1.05] | [0.91, 1.04] |
| | | | $P=7.05\times10^{-52}$ | $P=2.79\times10^{-40}$ |
| EMI | | | | −0.03 |
| | | | | [−0.08, 0.02] |
| | | | | $P=2.34\times10^{-1}$ |
| Intercept | −0.01 | 0.09 | −0.00 | 0.02 |
| | [−0.02, 0.01] | [0.00, 0.18] | [−0.03, 0.03] | [−0.03, 0.06] |
| | $P=2.92\times10^{-1}$ | $P=4.02\times10^{-2}$ | $P=9.92\times10^{-1}$ | $P=4.39\times10^{-1}$ |
| Observations | 71 | 71 | 71 | 71 |
| $R^2$ | 0.86 | 0.87 | 0.97 | 0.97 |
| Adjusted $R^2$ | 0.86 | 0.87 | 0.96 | 0.97 |
| $F$ statistic | 438.71 | 227.64 | 1,887.01 | 967.03 |
| | $P=1.28\times10^{-31}$ | $P=7.38\times10^{-31}$ | $P=7.52\times10^{-52}$ | $P=1.14\times10^{-50}$ |

Values in square brackets represent 95% CIs. Statistical significance was assessed using a two-sided $t$-test. No adjustments for multiple comparisons were made. Significant coefficients at the 0.05 level are in bold. $t$−1 refers to the preceding congressional session.

**EMI and inequality.** To measure income inequality, we use the share of pretax income of the top 1% of the population[28]. The data are from the world inequality database (https://wid.world/). A lagged correlation analysis shows that the strongest correlation between EMI and inequality has a lag 2, where EMI precedes inequality (Supplementary Fig. 6b). Inequality is also known to be correlated with polarization[18], which we also observe in our lagged correlation analysis in Supplementary Fig. 6c. For that reason, we study the role of EMI in inequality while considering polarization, as EMI and polarization are negatively cross-correlated. The VIF of a specification including lagged measures of inequality, EMI and polarization is 9.67, indicating that we need to include an interaction term between EMI and polarization. Thus, our model has the form

$$\mathrm{Ineq}(t) = a + b \times \mathrm{Ineq}(t-1) + c \times \mathrm{EMI}(t-1) + d \times \mathrm{Pol}(t-1)$$
$$+ e \times \mathrm{EMI}(t-1) \times \mathrm{Pol}(t-1).$$

We compare this model with a simple AR model including lagged values of inequality and polarization. The results are presented in Table 4. The lagged value of EMI has a negative and significant coefficient on inequality, and the interaction with polarization is not significant. Knowing the EMI in one session improves the prediction of inequality in the 2-year period that follows. The interaction between EMI and polarization, while positive, does not lead to an important mediation in the role of EMI, as shown in Supplementary Fig. 7a. Residuals in this model are stationary (ADF $P=0.022$, KSPP $P>0.1$) and do not deviate from normality (JB $P=0.822$).

The results of our analysis of inequality remain qualitatively similar with different specifications for the decisions we took in our analysis above. Supplementary Table 4 presents the results, where the first

model uses the Gini index as a measure of inequality. In this specification, the VIF of predictors is 13.17, indicating the need for inclusion of an interaction term between polarization and EMI. The coefficient of EMI is negative and significant, but it has a significant positive interaction with polarization. Supplementary Fig. 7b shows the shape of this interaction, revealing that high levels of polarization do not reverse the direction of association with EMI. Residuals in this model are stationary (ADF $P<0.01$, KPSS $P>0.1$) and do not deviate from normality (JB $P=0.343$). The second model uses the share of income of the top 1% of the population but includes less reliable data since 1912. In this model, the VIF is 3.175, but we keep the interaction term between polarization and EMI for comparability to other models. The result is similar as for the case using the Gini index: the coefficient for EMI is negative and significant, but the interaction with polarization is positive and significant. Supplementary Fig. 7c shows the shape of this interaction, where high levels of polarization do not reverse the slope of inequality with EMI. Residuals in this model are stationary (ADF $P<0.01$, KPSS $P=0.09$) but deviate from normality (JB $P<0.01$). For that reason, we performed a bootstrapping test on the coefficient of EMI with 10,000 samples, which indicates that the negative coefficient for EMI is robust to non-normal residuals (95% CI = [−0.34, −0.13]). The model with a longer lag for polarization also has a high VIF of 10.02, motivating the inclusion of the interaction between EMI and polarization. Residuals are stationary (ADF $P<0.01$, KPSS $P>0.1$) and do not deviate from normality (JB $P=0.716$). In this model, the coefficient of EMI is also negative and significant, and the interaction between polarization and EMI is significant only at the 0.1 level. Supplementary Fig. 7d shows the shape of this interaction, revealing the same pattern in which, even for high polarization, the slope of EMI is negative.

**Table 4 | Regression results of models of inequality (Ineq) as a function of lagged values of inequality, polarization, EMI and the interaction between EMI and polarization**

| | Ineq | |
|---|---|---|
| Ineq($t$−1) | **0.87** | **0.57** |
| | [0.70, 1.02] | [0.40, 0.74] |
| | $P=7.91\times10^{-13}$ | $P=6.67\times10^{-8}$ |
| Pol($t$−1) | **0.04** | 0.00 |
| | [0.00, 0.08] | [−0.04, 0.05] |
| | $P=4.69\times10^{-2}$ | $P=8.91\times10^{-1}$ |
| EMI($t$−1) | | **−0.11** |
| | | [−0.17, −0.05] |
| | | $P=1.18\times10^{-3}$ |
| Pol($t$−1)×EMI($t$−1) | | 0.08 |
| | | [−0.02, 0.18] |
| | | $P=1.28\times10^{-1}$ |
| Intercept | −0.01 | 0.06 |
| | [−0.02, 0.01] | [0.03, 0.10] |
| | $P=4.04\times10^{-1}$ | $P=1.45\times10^{-3}$ |
| Observations | 38 | 38 |
| $R^2$ | 0.92 | 0.95 |
| Adjusted $R^2$ | 0.92 | 0.95 |
| F statistic | 212.40 | 160.28 |
| | $P=2.67\times10^{-20}$ | $P=4.02\times10^{-21}$ |

Values in square brackets represent 95% CIs. Statistical significance was assessed using a two-sided $t$-test. No adjustments for multiple comparisons were made. Significant coefficients at the 0.05 level are in bold.

**Table 5 | Models of congressional productivity as a function of EMI and relevant covariates**

| | MLI | MLI | LPI | LPI | nlaw | nlaw |
|---|---|---|---|---|---|---|
| MLI(t−1) | **0.95** | **0.77** | | | | |
| | [0.82, 1.07] | [0.62, 0.92] | | | | |
| | $P=3.89×10^{-13}$ | $P=2.02×10^{-9}$ | | | | |
| Pol(t) | **−0.31** | 0.21 | **−0.45** | −0.14 | **−1.16** | **−0.89** |
| | [−0.49, −0.14] | [−0.29, 0.71] | [−0.79, −0.11] | [−0.47, 0.20] | [−1.45, −0.88] | [−1.43, −0.35] |
| | $P=1.17×10^{-3}$ | $P=3.84×10^{-1}$ | $P=1.18×10^{-2}$ | $P=4.04×10^{-1}$ | $P=3.52×10^{-9}$ | $P=2.18×10^{-3}$ |
| PartyControl(t) | −0.16 | −0.08 | −0.14 | −0.07 | −0.10 | −0.06 |
| | [−0.51, 0.20] | [−0.48, 0.31] | [−0.43, 0.15] | [−0.35, 0.22] | [−0.50, 0.30] | [−0.48, 0.35] |
| | $P=3.70×10^{-1}$ | $P=6.58×10^{-1}$ | $P=3.17×10^{-1}$ | $P=6.41×10^{-1}$ | $P=6.26×10^{-1}$ | $P=7.62×10^{-1}$ |
| PartyControlDif(t) | 0.19 | 0.27 | 0.23 | 0.29 | 0.01 | 0.05 |
| | [−0.22, 0.60] | [−0.14, 0.67] | [−0.15, 0.61] | [−0.00, 0.59] | [−0.32, 0.34] | [−0.26, 0.36] |
| | $P=3.44×10^{-1}$ | $P=1.87×10^{-1}$ | $P=2.24×10^{-1}$ | $P=5.04×10^{-2}$ | $P=9.41×10^{-1}$ | $P=7.36×10^{-1}$ |
| Mood(t) | **0.17** | **0.24** | **0.17** | **0.30** | **0.37** | **0.39** |
| | [0.01, 0.33] | [0.06, 0.41] | [0.04, 0.30] | [0.15, 0.44] | [0.19, 0.56] | [0.23, 0.56] |
| | $P=3.42×10^{-2}$ | $P=9.55×10^{-3}$ | $P=1.19×10^{-2}$ | $P=3.62×10^{-4}$ | $P=2.79×10^{-4}$ | $P=2.99×10^{-5}$ |
| EMI(t) | | **0.67** | | **0.83** | | 0.27 |
| | | [0.14, 1.20] | | [0.40, 1.26] | | [−0.05, 0.59] |
| | | $P=1.63×10^{-2}$ | | $P=7.24×10^{-4}$ | | $P=9.88×10^{-2}$ |
| EMI(t)×Pol(t) | | 0.23 | | 0.13 | | 0.00 |
| | | [−0.39, 0.85] | | [−0.39, 0.65] | | [−0.20, 0.20] |
| | | $P=4.44×10^{-1}$ | | $P=6.06×10^{-1}$ | | $P=9.80×10^{-1}$ |
| LPI(t−1) | | | **0.75** | **0.44** | | |
| | | | [0.57, 0.94] | [0.29, 0.60] | | |
| | | | $P=3.19×10^{-8}$ | $P=1.12×10^{-5}$ | | |
| nlaw(t−1) | | | | | −0.24 | −0.21 |
| | | | | | [−0.54, 0.06] | [−0.53, 0.12] |
| | | | | | $P=1.08×10^{-1}$ | $P=2.00×10^{-1}$ |
| Intercept | −0.16 | −0.14 | −0.25 | −0.43 | 0.03 | 0.01 |
| | [−0.40, 0.08] | [−0.47, 0.20] | [−0.48, −0.02] | [−0.70, −0.15] | [−0.15, 0.21] | [−0.25, 0.26] |
| | $P=1.82×10^{-1}$ | $P=4.05×10^{-1}$ | $P=3.27×10^{-2}$ | $P=3.98×10^{-3}$ | $P=7.31×10^{-1}$ | $P=9.64×10^{-1}$ |
| Observations | 27 | 27 | 27 | 27 | 36 | 36 |
| $R^2$ | 0.82 | 0.85 | 0.86 | 0.92 | 0.84 | 0.85 |
| Adjusted $R^2$ | 0.78 | 0.80 | 0.83 | 0.89 | 0.81 | 0.82 |
| F statistic | 18.90 | 15.45 | 25.67 | 30.91 | 31.58 | 23.06 |
| | $P=3.89×10^{-7}$ | $P=1.27×10^{-6}$ | $P=2.79×10^{-8}$ | $P=4.35×10^{-9}$ | $P=4.31×10^{-11}$ | $P=4.86×10^{-10}$ |

Fits start in 1951 to include policy mood data and end in 2004 for MLI and LPI and in 2022 for the logarithm of the number of laws (nlaw) passed in a session. Values in square brackets represent 95% CIs. Statistical significance was assessed using a two-sided t-test. No adjustments for multiple comparisons were made. Significant coefficients at the 0.05 level are in bold.

## Statistical analysis of the relationship between EMI and measures of congressional productivity

Following the specification of ref. 19, we fit a base model of three congressional productivity indices (MLI, LPI and log-transformed number of laws) as a function of the lagged dependent variable, polarization, policy mood and two indicator variables for whether the same party controls both the presidency and the majority in Congress and for a change in this variable. We extend this model by adding the EMI score of the same session in which productivity is measured. Thus, our model is for each variable Y (MLI, LPI and log laws)

$$Y(t) = a + b × Y(t-1) + c × Pol(t) + d × Mood(t) + e × PartyControl(t)$$
$$+ f × PartyControlDif(t) + g × EMI(t) + h × EMI(t) × Pol(t).$$

Note that, in this model specification, we use the EMI in the same session as the congressional productivity metric, as we aim to identify a correlation between variables that is robust to the known associations with other indicators. Table 5 presents regression results. Across the three models, explanatory variables reached VIF values up to 12.98, so we included an interaction term between polarization and EMI. Tests of stationarity of residuals had lower significance due to the smaller sample sizes (ADF $P = 0.26$ for MLI, $P = 0.07$ for LPI and $P = 0.03$ for number of laws), but KPSS tests were not significant in all three cases ($P > 0.1$) and JB tests were not significant either ($P > 0.5$). These small deviations from stationarity of residuals are corrected with the HAC covariance estimator.

While our analysis of productivity includes the important variable of mood, data on public policy mood are available only since the 1950s,

as they were collected via surveys. To analyse further the role of EMI in productivity, we adopt the approach of ref. 19, using the logarithm of the number of patents (from https://www.uspto.gov/web/offices/ac/ido/oeip/taf/h_counts.htm) approved during each session as an approximation of public mood regarding regulation. While this is an imperfect approximation, it allows us to study a much longer period, dating back to the nineteenth century. Thus, for each dependent variable, we now have models of the form

$$Y(t) = a + b \times Y(t-1) + c \times \text{Pol}(t) + d \times \text{npatents}(t)$$
$$+ e \times \text{PartyControl}(t) + f \times \text{PartyControlDif}(t),$$
$$+ g \times \text{EMI}(t) + h \times \text{EMI}(t) \times \text{Pol}(t)$$

where npatents($t$) represents the logarithm of the number of patents approved during the congressional session $t$. Covariates in this model have VIF up to 7.67 (LPI), and therefore we include an interaction term between EMI and Pol in each model. Results are presented in Supplementary Table 5. Residuals were approximately stationary, with significant ADF tests for MLI ($P = 0.014$) and number of laws ($P = 0.01$), and significant at the 0.1 level for LPI ($P = 0.09$). KPSS tests were not significant for all three models ($P > 0.1$), and JB tests were not significant for LPI ($P = 0.33$) and number of laws ($P = 0.52$). For MLI, a JB test was significant ($P < 0.01$), indicating non-normal residuals. For that reason, we performed a bootstrap test with 10,000 samples, which gave a 95% CI for the coefficient of EMI of [0.026, 0.214], indicating that the significant coefficient of the MLI model is robust to deviations from normality in the residuals. The coefficients of interaction terms between EMI and polarization are not significant and the coefficient for EMI is significant only for MLI, while it is not for LPI nor the number of laws.

### Reporting summary

Further information on research design is available in the Nature Portfolio Reporting Summary linked to this article.

## Data availability

Congressional speeches are available at https://data.stanford.edu/congress_text (ref. 39) and https://www.govinfo.gov/ (retrieved using https://github.com/unitedstates/congressional-record/). DW-NOMINATE scores are from https://voteview.com (ref. 26). Inequality data are from https://wid.world/. The Gini index is from https://www.census.gov/data/tables/time-series/demo/income-poverty/historical-income-inequality.html. Data on the number of patents are from https://www.uspto.gov/web/offices/ac/ido/oeip/taf/h_counts.htm. Data on public policy mood are available at https://stimson.web.unc.edu/data/ (ref. 29). Data on legislative productivity are available at https://doi.org/10.7910/DVN/ILILUD (ref. 19) and https://osf.io/mrghc/ (ref. 20). All the data used in this study are deposited in an Open Science Framework (OSF) repository at https://doi.org/10.17605/OSF.IO/Z6UTW (ref. 49).

## Code availability

Data were collected and analysed with Python (v3.6.13) and R (v4.3.1) scripts. The word embeddings models were trained using the implementation of Word2Vec algorithm in the Gensim library (v3.4.0). We efficiently apply the word embeddings using the sentence-transformers library (v2.2.2). The codes used to perform the analyses reported in this Article are available via GitHub at https://github.com/saroyehun/EvidenceMinusIntuition (with a snapshot available via Zenodo at https://doi.org/10.5281/zenodo.14288137 (ref. 50)).

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

## Acknowledgements

S.L. acknowledges financial support from the European Research Council (ERC Advanced Grant 101020961 PRODEMINFO), the Humboldt Foundation through a research award, the Volkswagen Foundation (grant 'Reclaiming individual autonomy and democratic discourse online: How to rebalance human and algorithmic decision making') and the European Commission (Horizon 2020 grant 101094752 SoMe4Dem). S.L. also receives funding from Jigsaw (a technology incubator created by Google) and from UK Research and Innovation through EU Horizon replacement funding grant number 10049415. D.G. is also a beneficiary of the ERC Advanced Grant 101020961 PRODEMINFO, and S.T.A. is supported by PRODEMINFO. D.G. also received funding from the Deutsche Forschungsgemeinschaft (DFG – German Research Foundation) under Germany's Excellence Strategy – EXC-2035/1 – 390681379. J.L. was supported by the Marie Skłodowska-Curie grant number 101026507. The funders had no role in the study design, data collection and analysis, decision to publish or preparation of the manuscript. We thank T. Brown for useful input on the manuscript.

## Author contributions

S.L., D.G. and S.T.A. conceptualized the research. S.T.A. collected the data. S.T.A. and D.G. developed the text analysis pipeline. F.C. performed the construction and validation of the keywords. A.S. performed the validation of the EMI score. D.G. and S.T.A. performed the statistical analyses. J.L. provided advice on the statistical analyses and visualization. S.L. and D.G. acquired funding and supervised the project. S.T.A., D.G. and S.L. prepared the initial draft of the manuscript. All authors contributed to preparing and editing the final version of the manuscript.

## Competing interests

The authors declare no competing interests.

## Additional information

**Correspondence and requests for materials** should be addressed to Stephan Lewandowsky.

# Reporting Summary

## Statistics

For all statistical analyses, confirm that the following items are present in the figure legend, table legend, main text, or Methods section.

| n/a | Confirmed | |
|---|---|---|
| ☐ | ☒ | The exact sample size (*n*) for each experimental group/condition, given as a discrete number and unit of measurement |
| ☐ | ☒ | A statement on whether measurements were taken from distinct samples or whether the same sample was measured repeatedly |
| ☐ | ☒ | The statistical test(s) used AND whether they are one- or two-sided *Only common tests should be described solely by name; describe more complex techniques in the Methods section.* |
| ☐ | ☒ | A description of all covariates tested |
| ☐ | ☒ | A description of any assumptions or corrections, such as tests of normality and adjustment for multiple comparisons |
| ☐ | ☒ | A full description of the statistical parameters including central tendency (e.g. means) or other basic estimates (e.g. regression coefficient) AND variation (e.g. standard deviation) or associated estimates of uncertainty (e.g. confidence intervals) |
| ☐ | ☒ | For null hypothesis testing, the test statistic (e.g. *F*, *t*, *r*) with confidence intervals, effect sizes, degrees of freedom and *P* value noted *Give P values as exact values whenever suitable.* |
| ☒ | ☐ | For Bayesian analysis, information on the choice of priors and Markov chain Monte Carlo settings |
| ☒ | ☐ | For hierarchical and complex designs, identification of the appropriate level for tests and full reporting of outcomes |
| ☒ | ☐ | Estimates of effect sizes (e.g. Cohen's *d*, Pearson's *r*), indicating how they were calculated |

*Our web collection on statistics for biologists contains articles on many of the points above.*

## Software and code

Policy information about availability of computer code

| Data collection | Data was collected with Python (v3.6.13) script. Specifically, the third-party Python package congressional-record (https://github.com/unitedstates/congressional-record) was used to update the existing collection of Congressional speeches. |
|---|---|
| Data analysis | Data was collected and analysed with Python (v3.6.13) and R (v4.3.1) scripts. The word embeddings models were trained using the implementation of Word2Vec algorithm in the Gensim library (v3.4.0). We efficiently apply the word embeddings using sentence-transformers library (v2.2.2). Codes for data analysis are available in a Github repository (https://github.com/saroyehun/EvidenceMinusIntuition) with a snapshot at https://doi.org/10.5281/zenodo.14288137. |

For manuscripts utilizing custom algorithms or software that are central to the research but not yet described in published literature, software must be made available to editors and reviewers. We strongly encourage code deposition in a community repository (e.g. GitHub). See the Nature Portfolio guidelines for submitting code & software for further information.

## Data

Policy information about availability of data

All manuscripts must include a data availability statement. This statement should provide the following information, where applicable:

- Accession codes, unique identifiers, or web links for publicly available datasets
- A description of any restrictions on data availability
- For clinical datasets or third party data, please ensure that the statement adheres to our policy

> Congressional speeches are available from https://data.stanford.edu/congress_text and https://www.govinfo.gov(retrieved using https://github.com/unitedstates/congressional-record/). DW-NOMINATE scores are from https://voteview.com. Inequality data are from https://wid.world/. Data on number of patents are from https://www.uspto.gov/web/offices/ac/ido/oeip/taf/h_counts.htm. Data on public policy mood are available from https://stimson.web.unc.edu/data/. Data on legislative productivity are available from https://doi.org/10.7910/DVN/ILILUD and https://osf.io/mrghc. All the data used in this study are deposited in an Open Science Framework (OSF) repository (https://doi.org/10.17605/OSF.IO/Z6UTW).

## Research involving human participants, their data, or biological material

Policy information about studies with human participants or human data. See also policy information about sex, gender (identity/presentation), and sexual orientation and race, ethnicity and racism.

| | |
|---|---|
| Reporting on sex and gender | For the validation of dictionaries and the EMI score, participants were recruited through Prolific. For these validation tasks, we were not concerned with analysis based on sex or gender and did not directly collect this information. The validation analyses aimed solely to obtain assessments from a sample broadly representative of the U.S. population. However, information available on our sample from Prolific indicates the following:<br>- For dictionary validation, participants in our sample self-identified as 19 male, 30 female, and 1 non-binary.<br>- For EMI score validation, participants in our sample self-identified as 80 female, 72 male, 3 preferred not to say, and 1 had missing data. |
| Reporting on race, ethnicity, or other socially relevant groupings | We did not collect information on race, ethnicity, or other socially relevant groupings. |
| Population characteristics | For the validation of the computational text analysis: representative sample of adults in the United States. |
| Recruitment | For the validation of the computational text analysis: Prolific |
| Ethics oversight | The dictionary validation task was carried out under Ethics approval from the University of Bristol (ethics application #12299). For the validation of the EMI score, the University of Konstanz ethics review board granted an ethics review exemption for the annotation task used to validate the EMI score. |

Note that full information on the approval of the study protocol must also be provided in the manuscript.

# Field-specific reporting

Please select the one below that is the best fit for your research. If you are not sure, read the appropriate sections before making your selection.

☐ Life sciences  ☒ Behavioural & social sciences  ☐ Ecological, evolutionary & environmental sciences

For a reference copy of the document with all sections, see nature.com/documents/nr-reporting-summary-flat.pdf

# Behavioural & social sciences study design

All studies must disclose on these points even when the disclosure is negative.

| | |
|---|---|
| Study description | Quantitative text analysis of US Congressional Speeches from 1879 to 2022. |
| Research sample | The speeches of U.S. Congress Members that were analysed in this study were compiled from publicly available sources, mainly https://www.govinfo.gov/ and https://data.stanford.edu/congress_text. The dataset includes speeches from both the House of Representatives and the Senate, spanning the period from 1879 to 2022. Demographic attributes such as age and sex were not directly available for all records. However, given that the dataset encompasses all transcribed speeches from U.S. Congress members during this period, it provides a comprehensive representation of Congressional discourse over time. The selection of this dataset ensures that the analyses captures the full range of legislative rhetoric over time without potential bias introduced by sampling constraints. |
| Sampling strategy | The speeches included in this study are a comprehensive collection of transcribed speeches made on the floor of US Congress from 1879 to 2022. No sampling was involved in the compilation of the data and no sample size calculation was performed. Given the comprehensive nature of the dataset, its size is sufficient to support robust analyses without the need for sampling. |

| | |
|---|---|
| Data collection | Datasets were retrieved through download and automatic parsing of U.S. Congressional records. Since data collection involved no direct researcher-participant interaction, no additional individuals were present. As the study is observational and based on archival data, researcher blinding to experimental conditions and hypotheses does not apply. |
| Timing | The data spans Congressional sessions from January 1879 to December 2022. |
| Data exclusions | We excluded a total of 7,341,187 entries, including 7,336,909 duplicates and procedural speeches, as well as 4,278 noisy text (such as those resulting from Optical Character Recognition rendering of tables). We included only speeches from Democrat and Republican Congress Members. |
| Non-participation | There are no known cases of non-participation as this is an observational study. |
| Randomization | We did not perform randomization because this study is purely observational. |

# Reporting for specific materials, systems and methods

We require information from authors about some types of materials, experimental systems and methods used in many studies. Here, indicate whether each material, system or method listed is relevant to your study. If you are not sure if a list item applies to your research, read the appropriate section before selecting a response.

## Materials & experimental systems

| n/a | Involved in the study |
|---|---|
| ☒ | Antibodies |
| ☒ | Eukaryotic cell lines |
| ☒ | Palaeontology and archaeology |
| ☒ | Animals and other organisms |
| ☒ | Clinical data |
| ☒ | Dual use research of concern |
| ☒ | Plants |

## Methods

| n/a | Involved in the study |
|---|---|
| ☒ | ChIP-seq |
| ☒ | Flow cytometry |
| ☒ | MRI-based neuroimaging |

## Plants

| | |
|---|---|
| Seed stocks | N/A |
| Novel plant genotypes | N/A |
| Authentication | N/A |

