## [Peer Review File · Nature Human Behaviour]

Computational analysis of US Congressional speeches reveals a shift from evidence to intuition

Corresponding Author: Professor Stephan Lewandowsky

Version 0:

Decision Letter:

15th October 2024

Dear Professor Lewandowsky,

Thank you once again for your manuscript, entitled "Computational analysis of 145 years of US Congressional speeches reveals shift from evidence to intuition," and for your patience during the peer review process.

Your manuscript has now been evaluated by 2 reviewers, whose comments are included at the end of this letter. A third reviewer had to be withdrawn. Although the reviewers find your work to be of interest, they also raise some important concerns. We are very interested in the possibility of publishing your study in Nature Human Behaviour, but would like to consider your response to these concerns in the form of a revised manuscript before we make a decision on publication.

To guide the scope of the revisions, the editors discuss the referee reports in detail within the team, including with the chief editor, with a view to (1) identifying key priorities that should be addressed in revision and (2) overruling referee requests that are deemed beyond the scope of the current study. We hope that you will find the prioritised set of referee points to be useful when revising your study. Please do not hesitate to get in touch if you would like to discuss these issues further.

1. Both reviewers ask that you consider language change over time in your analyses. Please address this concern in full and provide additional analyses on the impact of semantic change on your results.

2. Reviewer 3 raises concerns that the topic of the speech could be affecting the results. Please provide additional analyses to alleviate these concerns.

In sum, we invite you to revise your manuscript taking into account all reviewer and editor comments. We are committed to providing a fair and constructive peer-review process. Do not hesitate to contact us if there are specific requests from the reviewers that you believe are technically impossible or unlikely to yield a meaningful outcome.

We hope to receive your revised manuscript within two months. I would be grateful if you could contact us as soon as possible if you foresee difficulties with meeting this target resubmission date.

- Include a "Response to the editors and reviewers" document detailing, point-by-point, how you addressed each editor and referee comment. If no action was taken to address a point, you must provide a compelling argument. When formatting this document, please respond to each reviewer comment individually, including the full text of the reviewer comment verbatim followed by your response to the individual point. This response will be used by the editors to evaluate your revision and sent back to the reviewers along with the revised manuscript.

- Highlight all changes made to your manuscript or provide us with a version that tracks changes.

Link Redacted

Note: This URL links to your confidential home page and associated information about manuscripts you may have submitted, or

that you are reviewing for us. If you wish to forward this email to co-authors, please delete the link to your homepage.

We look forward to seeing the revised manuscript and thank you for the opportunity to review your work. Please do not hesitate to contact me if you have any questions or would like to discuss these revisions further.

Sincerely,

[REDACTED]
[REDACTED]
[REDACTED]
Nature Human Behaviour

Reviewer expertise:

Reviewer #1: NLP ; CSS ; political communication

Reviewer #2: NLP ; communication *withdrawn*

Reviewer #3: NLP ; political communication

REVIEWER COMMENTS:

Reviewer #1 (Remarks to the Author):

Key results: Please summarise what you consider to be the outstanding features of the work.

Answer:

- (1) long time span,
- (2) important research questions of our contemporary society (post-truth era, polarization, shrinking public sphere, loss of trust in institutions)
- (3) validated analyses from multiple perspectives
- (4) To my knowledge, the corpus (and political language use in general) has not been studied with such a research question ("how the search for truth is reflected in political rhetoric").

Validity: Does the manuscript have flaws which should prohibit its publication? If so, please provide details.

Answer:

No. The indicators and the models are statistically well chosen. More indicators, more statistical models validate the robustness of the results.

Originality and significance: What are the major claims of the paper? Do you think that they represent a significant advance in the field? If the conclusions are not original, please provide relevant references. On a more subjective note, do you feel that the results presented are of immediate interest to many people in your own discipline, and/or to people from several disciplines?

Answer:

The conclusions are original and represent a significant advance in the field. I believe that these results will be of direct interest to many people in sociology and other social sciences.

Conceptual novelty: (1) The paper adopts a framework that distinguishes between two different ways (evidence-based and intuition-based) in which people can express their search for truth. (2) It defines further key indicators related to the functioning of the democratic public sphere (political polarization, income inequality, congressional productivity).

Methodological novelty: (1) Measuring the linguistic imprint of these two approaches (evidence and intuition-based reasoning) in political discourse (2) Measuring their interrelationship over a 150-year period.

Evidence-based societal progress: (1) evidence-based language has continued to decline since the mid-1970s (2) a time-shifted correlation of the above indicators, suggesting possible causal relationships (e.g., greater emphasis on evidence-based reasoning is followed by reduced income inequality; evidence-based language has a positive impact on congressional productivity). (3) These potential causal relationships underscore the importance of evidence-based decision making.

Data & methodology: Please comment on the validity of the approach, quality of the data and quality of presentation. Please note that we expect our reviewers to review all data, including any extended data and supplementary information. Is the reporting of data and methodology sufficiently detailed and transparent to enable reproducing the results?

Answer:

Yes, the reporting of data and methodology sufficiently detailed and transparent to enable reproducing the results. The data and codes are publicly available.

Preregistration: If any part of the work reported in the manuscript was pre-registered, did the authors follow their preregistration plan?

Answer:

Not applicable: the analysis was not pre-registered

Appropriate use of statistics and treatment of uncertainties: Please include in your report a specific comment on the appropriateness of any statistical tests, and the accuracy of the description of any error bars and probability values.

Answer:

All the tests and statistical analysis are appropriate.

Conclusions: Do you find that the conclusions and data interpretation are robust, valid and reliable?

Answer:

Yes, conclusions and data interpretation are robust, valid and reliable.

Suggested improvements:

150 years have changed language, spelling and vocabulary. Does this not pose a methodological problem? For example, did you try to take this into account in the preprocessing, in the standardization of the different word forms, or in the definition of the keyword list, and not only use today's American English as a starting point, but also take samples from older texts? (I see that the validation of the EMI score is based on a qualitative text analysis of a sample of the whole corpus).

References: Does this manuscript reference previous literature appropriately? If not, what references should be included or excluded?

Answer:

Please point out in the paper that the political polarization indicator used (dw-nominate) measures only one specific definition of polarization (estimates political positions for legislators from legislative votes), and that there are many other definitions and indicators, including ones based on computational text analysis. Polarization can be measured as the dispersion of opinions, as the bi-modality of opinions, as the correlation between social attitudes and salient individual characteristics, as perceived polarization, as affective polarization or as interactional polarization etc.

The indicator used here is widely used in political science but does not necessarily correspond to the reader's or the public's concept of polarization, which is why this clarification is important.

Clarity and context: Is the abstract clear, accessible? Are abstract, introduction and conclusions appropriate?

Answer:

The whole paper is very clear and easy to follow.

If applicable, Remarks on Code Availability (which are transmitted in full)

Codes used for data collection and analysis are available on github.

Reviewer #3 (Remarks to the Author):

The authors present a computational look at the evolution of congressional arguments to show the reliance on evidence versus intuition. The study addresses an important question with practical implications for rhetoric and policy. The dataset is larger than many others in this domain and the analysis is appropriate for answering the central question. While the authors do make a valiant attempt with the supplemental materials to rule out alternative explanations and confounding factors, a few major issues are not considered sufficiently.

First, there doesn't seem to be a consideration of language change over time. The embeddings model is built from the entire corpus and only one set of dictionaries. Further analysis is needed to show the potential impact (or not) of semantic change for those words over the time period. The human validation component addresses this somewhat, but I do not believe it goes far enough in showing that the semantic usages of the words stay relatively constant. Crowdsourcing doesn't seem like the best validation here; political historians may have a needed insight into the evaluating the older text in context. Another method of validation would be to track the changing is the word embedding over time for at least several of the keywords; this would address another potential concern which is that the embedding model is skewed toward the more modern period (a graph of the number of speeches by year could show if this is worth testing).

Second, the topic of the speech could be affecting the results. For example, evidence-based language could be more common in speeches about the economy while intuition-based language is more common in speeches about social issues. The distribution of topics in congressional has likely changed significantly over time in ways that change the likelihood of different types of arguments. Analyzing subsets of the speeches separately by policy topic to see if the same patterns emerge would be a potential way to address this issue.

Finally, though somewhat addressed in the discussion, I would like to see the results further contextualized with the history and structure of congressional rules. The authors are clear in stating this is a complex issue with many factors such as the role of party leadership. However, some discussion should be some rules and factors changes over this time period (i.e. party realignment in the 1960s; gradual loss of power to the executive branch over the 20th century; changes to the rules around length of debates). A few minor points for change would be more discussion of how the dictionaries were developed. The choice for a single index rather than two separate metrics could be justified more. The analysis related to Figure 1 could be expanded to find the inflection point in the regression line to more clearly define the shift in style.

Overall, however, this is a very interesting and well-done piece of research.

Version 1:

Decision Letter:

Our ref: NATHUMBEHAV-24062375A

2nd January 2025

Dear Dr. Lewandowsky,

Thank you for submitting your revised manuscript "Computational analysis of 145 years of US Congressional speeches reveals shift from evidence to intuition" (NATHUMBEHAV-24062375A). It has now been seen by the original referees and their comments are below. As you can see, the reviewers find that the paper has improved in revision. We will therefore be happy in principle to publish it in Nature Human Behaviour, pending minor revisions to satisfy the referees' final requests and to comply with our editorial and formatting guidelines.

We are now performing detailed checks on your paper and will send you a checklist detailing our editorial and formatting requirements within two weeks. Please do not upload the final materials and make any revisions until you receive this additional information from us.

Sincerely,

[REDACTED]
[REDACTED]
[REDACTED]
[REDACTED]
Nature Human Behaviour

Reviewer #1 (Remarks to the Author):

The authors have adequately responded to all criticisms raised. I recommend the revised paper for publication.

Reviewer #1 (Remarks on code availability):

Verifying the Python code is outside my area of expertise.

Reviewer #3 (Remarks to the Author):

The authors sufficiently addressed my concerns.

Reviewer #3 (Remarks on code availability):

I briefly reviewed the github repo though I did not run any of the code. The README seems clear and the code well-structured however.

Version 2:

Decision Letter:

Dear Professor Lewandowsky,

We are pleased to inform you that your Article "Computational analysis of US Congressional speeches reveals a shift from evidence to intuition", has now been accepted for publication in Nature Human Behaviour.

Once your manuscript is typeset and you have completed the appropriate grant of rights, you will receive a link to your electronic proof via email with a request to make any corrections within 48 hours. If, when you receive your proof, you cannot meet this deadline, please inform us at rjsproduction@springernature.com immediately. Once your paper has been scheduled for online

publication, the Nature press office will be in touch to confirm the details.

With best regards,

Nature Human Behaviour

P.S. Click on the following link if you would like to recommend Nature Human Behaviour to your librarian <http://www.nature.com/subscriptions/recommend.html#forms>

** Visit the Springer Nature Editorial and Publishing website at http://editorial-jobs.springernature.com?utm_source=ejp_NHumB_email&utm_medium=ejp_NHumB_email&utm_campaign=ejp_NHumB for more information about our career opportunities. If you have any questions please click [here](mailto:editorial.publishing.jobs@springernature.com).

Open Access This Peer Review File is licensed under a Creative Commons Attribution 4.0 International License, which permits use, sharing, adaptation, distribution and reproduction in any medium or format, as long as you give appropriate credit to the original author(s) and the source, provide a link to the Creative Commons license, and indicate if changes were made. In cases where reviewers are anonymous, credit should be given to 'Anonymous Referee' and the source. The images or other third party material in this Peer Review File are included in the article's Creative Commons license, unless indicated otherwise in a credit line to the material. If material is not included in the article's Creative Commons license and your intended use is not permitted by statutory regulation or exceeds the permitted use, you will need to obtain permission directly from the copyright holder.

Response to Reviewers - Computational analysis of 145 years of US Congressional speeches reveals shift from evidence to intuition

>> We would like to thank all the reviewers for their time and effort in providing feedback on our manuscript. Below, we provide responses to all the points raised.

REVIEWER COMMENTS:

Reviewer #1 (Remarks to the Author):

Key results: Please summarise what you consider to be the outstanding features of the work.

Answer:

- (1) long time span,
- (2) important research questions of our contemporary society (post-truth era, polarization, shrinking public sphere, loss of trust in institutions)
- (3) validated analyses from multiple perspectives
- (4) To my knowledge, the corpus (and political language use in general) has not been studied with such a research question ("how the search for truth is reflected in political rhetoric").

Validity: Does the manuscript have flaws which should prohibit its publication? If so, please provide details.

Answer:

No. The indicators and the models are statistically well chosen. More indicators, more statistical models validate the robustness of the results.

Originality and significance: What are the major claims of the paper? Do you think that they represent a significant advance in the field? If the conclusions are not original, please provide relevant references. On a more subjective note, do you feel that the results presented are of immediate interest to many people in your own discipline, and/or to people from several disciplines?

Answer:

The conclusions are original and represent a significant advance in the field. I believe that these results will be of direct interest to many people in sociology and other social sciences.

Conceptual novelty: (1) The paper adopts a framework that distinguishes between two different ways (evidence-based and intuition-based) in which people can express their search for truth.

(2) It defines further key indicators related to the functioning of the democratic public sphere (political polarization, income inequality, congressional productivity).

Methodological novelty: (1) Measuring the linguistic imprint of these two approaches (evidence and intuition-based reasoning) in political discourse (2) Measuring their interrelationship over a 150-year period.

Evidence-based societal progress: (1) evidence-based language has continued to decline since the mid-1970s (2) a time-shifted correlation of the above indicators, suggesting possible causal relationships (e.g., greater emphasis on evidence-based reasoning is followed by reduced

income inequality; evidence-based language has a positive impact on congressional productivity). (3) These potential causal relationships underscore the importance of evidence-based decision making.

>> Thank you for highlighting the strengths of our work.

Data & methodology: Please comment on the validity of the approach, quality of the data and quality of presentation. Please note that we expect our reviewers to review all data, including any extended data and supplementary information. Is the reporting of data and methodology sufficiently detailed and transparent to enable reproducing the results?

Answer:

Yes, the reporting of data and methodology sufficiently detailed and transparent to enable reproducing the results. The data and codes are publicly available.

Preregistration: If any part of the work reported in the manuscript was pre-registered, did the authors follow their preregistration plan?

Answer:

Not applicable: the analysis was not pre-registered

Appropriate use of statistics and treatment of uncertainties: Please include in your report a specific comment on the appropriateness of any statistical tests, and the accuracy of the description of any error bars and probability values.

Answer:

All the tests and statistical analysis are appropriate.

Conclusions: Do you find that the conclusions and data interpretation are robust, valid and reliable?

Answer:

Yes, conclusions and data interpretation are robust, valid and reliable.

Suggested improvements:

150 years have changed language, spelling and vocabulary. Does this not pose a methodological problem? For example, did you try to take this into account in the preprocessing, in the standardization of the different word forms, or in the definition of the keyword list, and not only use today's American English as a starting point, but also take samples from older texts? (I see that the validation of the EMI score is based on a qualitative text analysis of a sample of the whole corpus).

>>This is a valid point. We address this concern with a new set of analyses following the suggestion by R3 (see below).

Although we conducted validation on a sample of the dataset, we were careful to ensure that the text samples were stratified over time. Furthermore, since we rely on word embeddings, potential variations of the same word which are likely to appear in similar contexts will be captured by the embeddings model (see <https://doi.org/10.18653/v1/2020.coling-main.75>). For

example, an inspection of the nearest neighbours of the word 'evidence' include the word 'evilence' which is a spelling mistake that likely stems from OCR error which the embeddings model still captures. Hence, such variations are covered in our computational text analysis method.

References: Does this manuscript reference previous literature appropriately? If not, what references should be included or excluded?

Answer:

Please point out in the paper that the political polarization indicator used (dw-nominate) measures only one specific definition of polarization (estimates political positions for legislators from legislative votes), and that there are many other definitions and indicators, including ones based on computational text analysis. Polarization can be measured as the dispersion of opinions, as the bi-modality of opinions, as the correlation between social attitudes and salient individual characteristics, as perceived polarization, as affective polarization or as interactional polarization etc.

The indicator used here is widely used in political science but does not necessarily correspond to the reader's or the public's concept of polarization, which is why this clarification is important.

>> We have included a clarification on this point. We say on pg. 5 in the main text of the revised manuscript:

"It is important to clarify that the political polarization indicator used in this study, DW-NOMINATE, measures polarization using voting behaviour within a legislative context. However, polarization is a complex and multifaceted concept with various definitions and indicators, including affective polarization, issue polarization, and perceived polarization. Additional indicators can also be derived from computational text analysis, as well as from opinion, structural, and interactional dynamics. Exploring these alternative measures is beyond the scope of the current study."

Clarity and context: Is the abstract clear, accessible? Are abstract, introduction and conclusions appropriate?

Answer:

The whole paper is very clear and easy to follow.

If applicable, Remarks on Code Availability (which are transmitted in full)

Codes used for data collection and analysis are available on github.

Reviewer #3 (Remarks to the Author):

The authors present a computational look at the evolution of congressional arguments to show the reliance on evidence versus intuition. The study addresses an important question with practical implications for rhetoric and policy. The dataset is larger than many others in this domain and the analysis is appropriate for answering the central question. While the authors do

make a valiant attempt with the supplemental materials to rule out alternative explanations and confounding factors, a few major issues are not considered sufficiently.

First, there doesn't seem to be a consideration of language change over time. The embeddings model is built from the entire corpus and only one set of dictionaries. Further analysis is needed to show the potential impact (or not) of semantic change for those words over the time period.

The human validation component addresses this somewhat, but I do not believe it goes far enough in showing that the semantic usages of the words stay relatively constant.

Crowdsourcing doesn't seem like the best validation here; political historians may have a needed insight into the evaluating the older text in context. Another method of validation would be to track the changing is the word embedding over time for at least several of the keywords; this would address another potential concern which is that the embedding model is skewed toward the more modern period (a graph of the number of speeches by year could show if this is worth testing).

>>This is a very important point. To examine the potential impact of language change, we now also train temporal embeddings models on splits of the corpus into 2-decade slices. To ensure comparability, we downsample recent periods to achieve a comparable number of tokens (about 100M) across all temporal slices.

To quantify the consistency of meaning for the keywords in our analysis, we compute the average pairwise cosine similarity between each dictionary word across all combinations of time periods (i.e., slices of two decades each) using the respective temporal embeddings. As a baseline, we calculate the average similarity of each dictionary word with a randomly selected word. The results for both the evidence (panel A below) and intuition (in the Supplementary Notes S11) dictionaries show higher average similarities compared to the baseline (panel B below). While there is some variation in the average similarity across time slices, it remains substantially higher than those observed in the baseline. The results demonstrate the temporal stability of the keywords.

We also compute the EMI score using temporal embeddings. Specifically, we apply embeddings for a given time period to documents from the same time period and the dictionary. The results indicate that the trend of the EMI score is very similar to the one computed using an embeddings model trained on the full corpus, in particular the downward trend from the mid-1970s.

We include the results of the analyses using temporal embeddings in Supplementary Notes S11.

Second, the topic of the speech could be affecting the results. For example, evidence-based language could be more common in speeches about the economy while intuition-based language is more common in speeches about social issues. The distribution of topics in congressional has likely changed significantly over time in ways that change the likelihood of different types of arguments. Analyzing subsets of the speeches separately by policy topic to see if the same patterns emerge would be a potential way to address this issue.

>> This is an excellent point. To address this concern, we train a classifier (based on transformers) on the CAP (Comparative Agendas Project) dataset covering 20 policy areas and a non-policy category (Others). The classifier predicts the most likely policy area covered by each piece of text. As suggested, we plot the EMI score over time for each policy area (normalizing within each topic). The results show very similar trends across all policy areas (and a non-policy category, 'Others'). Also, the macro average over the EMI score for each topic reflects the trend that we observed. We report the results on topic analysis in the Supplementary Notes S10 (also see figure below).

Finally, though somewhat addressed in the discussion, I would like to see the results further contextualized with the history and structure of congressional rules. The authors are clear in stating this is a complex issue with many factors such as the role of party leadership. However,

some discussion should be some rules and factors changes over this time period (i.e. party realignment in the 1960s; gradual loss of power to the executive branch over the 20th century; changes to the rules around length of debates).

>> Thank you for your suggestion. We now include these additional points in the Discussion section (on pg. 9). Specifically, we say:

“Modifications to Congressional rules and procedures, particularly around the length of debates, can influence the breadth and depth of discussions on the Congressional floor. For example, the introduction of the “cloture” rule in the Senate in 1917 provided a mechanism to limit debate time and expedite legislative processes. Before this, there was no formal method to end a debate or force a vote on an issue, which allowed for extended deliberations. While such rules may improve efficiency, they can also shorten discussions and potentially limit the richness of legislative debates. The evolving nature of Congressional rules and procedures can influence the characteristics of discourse on the Congressional floor over time.

Presidents have increasingly sought to expand their powers, often justified by their role as commander-in-chief, particularly during crises or in an attempt to unilaterally advance their policy agendas [35]. Mechanisms such as executive orders and the creation of administrative agencies under presidential control have facilitated this expansion. While some of these actions are supported by Congressional authorization, the steady accumulation of executive power may have implications for the legislative branch. This expansion may limit the sphere of influence of Congress, potentially reducing its role to rubber-stamping presidential initiatives. Conversely, it can also lead to tensions and heightened oversight efforts by Congress on activities of the executive branch and agencies. The balance of power between the executive and legislative branches can shape the nature and focus of Congressional discourse.”

A few minor points for change would be more discussion of how the dictionaries were developed. The choice for a single index rather than two separate metrics could be justified more.

>>We now provide more details on the construction of the dictionary in the Methods section. The reasoning behind using a single index is to capture the relative emphasis on evidence-based and intuition-based language in a piece of text. This approach reflects our conception of Truth as a continuum, where the relative salience of these two forms of reasoning provides greater insight than considering each one of them independently. By combining the scores into a single index, we aim to achieve a better operationalization of this conceptual framework. For completeness, we also include a plot in Supplementary Figure 2 (also see figure below) showing the separate evidence-based and intuition-based scores over time.

The analysis related to Figure 1 could be expanded to find the inflection point in the regression line to more clearly define the shift in style.

>> This is a great suggestion. Accordingly, we fit a Multivariate Adaptive Regression Splines (MARS) model in Python. We apply this model to versions of the EMI time series. The results (see Supplementary Notes S12) indicate a breakpoint in the session 1973-74 (with the second highest EMI score), which is very close to the peak session of 1975-76 identified in our initial analysis.

Overall, however, this is a very interesting and well-done piece of research.

>>Thank you very much. We are very happy to hear that.